# Stochastic Gradient Geodesic MCMC Methods

**Chang Liu[†], Jun Zhu[†], Yang Song[‡*]**
[†] Dept. of Comp. Sci. & Tech., TNList Lab; Center for Bio-Inspired Computing Research
[†] State Key Lab for Intell. Tech. & Systems, Tsinghua University, Beijing, China
[‡] Dept. of Physics, Tsinghua University, Beijing, China
{chang-li14@mails, dcszj@}.tsinghua.edu.cn; songyang@stanford.edu

## Abstract

We propose two stochastic gradient MCMC methods for sampling from Bayesian posterior distributions defined on Riemann manifolds with a known geodesic flow, e.g. hyperspheres. Our methods are the first scalable sampling methods on these manifolds, with the aid of stochastic gradients. Novel dynamics are conceived and 2nd-order integrators are developed. By adopting embedding techniques and the geodesic integrator, the methods do not require a global coordinate system of the manifold and do not involve inner iterations. Synthetic experiments show the validity of the method, and its application to the challenging inference for spherical topic models indicate practical usability and efficiency.

## 1 Introduction

Dynamics-based Markov Chain Monte Carlo methods (D-MCMCs) are sampling methods using dynamics simulation for state transition in a Markov chain. They have become a workhorse for Bayesian inference, with well-known examples like Hamiltonian Monte Carlo (HMC) [22] and stochastic gradient Langevin dynamics (SGLD) [29]. Here we consider variants for sampling from distributions defined on Riemann manifolds. Overall, geodesic Monte Carlo (GMC) [7] stands out for its notable performance on manifolds with known geodesic flow, such as simplex, hypersphere and Stiefel manifold [26, 16]. Its applicability to manifolds with no global coordinate systems (e.g. hyperspheres) is enabled by the embedding technique, and its geodesic integrator eliminates inner (within one step in dynamics simulation) iteration to ensure efficiency. It is also used for efficient sampling from constraint distributions [17]. Constrained HMC (CHMC) [6] aims at manifolds defined by a constraint in some $\mathbb{R}^n$. It covers all common manifolds, but inner iteration makes it less appealing. Other D-MCMCs involving Riemann manifold, e.g. Riemann manifold Langevin dynamics (RMLD) and Riemann manifold HMC (RMHMC) [13], are invented for better performance but still on the task of sampling in Euclidean space, where the target variable is treated as the global coordinates of some distribution manifold. Although they can be used to sample in non-Euclidean Riemann manifolds by replacing the distribution manifold with the target manifold, a global coordinate system of the target manifold is required. Moreover, RMHMC suffers from expensive inner iteration.

However, GMC scales undesirably to large datasets, which are becoming common. An effective strategy to scale up D-MCMCs is by randomly sampling a subset to estimate a noisy but unbiased stochastic gradient, with stochastic gradient MCMC methods (SG-MCMCs). Welling et al. [29] pioneered in this direction by developing stochastic gradient Langevin dynamics (SGLD). Chen et al. [9] apply the idea to HMC with stochastic gradient HMC (SGHMC), where a non-trivial dynamics with friction has to be conceived. Ding et at. [10] propose stochastic gradient Nosé-Hoover thermostats (SGNHT) to automatically adapt the friction to the noise by a thermostats. To unify dynamics used for SG-MCMCs, Ma et al. [19] develop a complete recipe to formulate the dynamics.

---

[*]JZ is the corresponding author; YS is with Department of Computer Science, Stanford University, CA.

Table 1: A summary of some D-MCMCs. –: sampling on manifold not supported; †: The integrators are not in the SSI scheme (It is unclear whether the claimed "2nd-order" is equivalent to ours); ‡: 2nd-order integrators for SGHMC and mSGNHT are developed by [8] and [18], respectively.

| methods | stochastic gradient | no inner iteration | no global coordinates | order of integrator |
|---|---|---|---|---|
| GMC [7] | $\times$ | $\checkmark$ | $\checkmark$ | 2nd |
| RMLD [13] | $\times$ | $\checkmark$ | $\times$ | 1st |
| RMHMC [13] | $\times$ | $\times$ | $\times$ | 2nd$^\dagger$ |
| CHMC [6] | $\times$ | $\times$ | $\checkmark$ | 2nd$^\dagger$ |
| SGLD [29] | $\checkmark$ | $\checkmark$ | – | 1st |
| SGHMC [9] / SGNHT [10] | $\checkmark$ | $\checkmark$ | – | 1st$^\ddagger$ |
| SGRLD [23] / SGRHMC [19] | $\checkmark$ | $\checkmark$ | $\times$ | 1st |
| SGGMC / gSGNHT (proposed) | $\checkmark$ | $\checkmark$ | $\checkmark$ | 2nd |

In this paper, we present two SG-MCMCs for manifolds with known geodesic flow: stochastic gradient geodesic Monte Carlo (SGGMC) and geodesic stochastic gradient Nosé-Hoover thermostats (gSGNHT). They are the first scalable sampling methods on manifolds with known geodesic flow and no global coordinate systems. We use the recipe [19] to tackle the non-trivial dynamics conceiving task. Our novel dynamics are also suitable for developing 2nd-order integrators by adopting the *symmetric splitting integrator* (SSI) [8] scheme. A key property of a $K$th-order integrator is the bias of the expected sample average at iteration $L$ can be upper bounded by $L^{-K/(K+1)}$ and the mean square error by $L^{-2K/(2K+1)}$ [8], so a higher order integrator basically performs better. Our integrators also incorporate the geodesic integrator to avoid inner iteration. Our methods can also be used to scalably sample from constraint distributions [17] like GMC.

There exist other SG-MCMCs on Riemann manifold, e.g. SGRLD [23] and SGRHMC [19], stochastic gradient versions of RMLD and RMHMC respectively. But they also require the Riemann manifold to have a global coordinate system, like their original versions as is mentioned above. So basically they cannot draw samples from hyperspheres, while our methods are capable. Technically, SGRLD/SGRHMC (and RMLD/RMHMC) samples in the coordinate space, so we need a global one to make it valid. The explicit use of the Riemann metric tensor also makes the methods more difficult to implement. Our methods (and GMC) sample in the isometrically embedded space, where the whole manifold is represented and the Riemann metric tensor is implicitly embodied by the isometric embedding. Moreover, our integrators are of a higher order. Tab. 1 summarizes the key properties of aforementioned D-MCMCs, where our advantages are clearly shown.

Finally, we apply our samplers to perform inference for spherical admixture models (SAM) [24]. SAM defines a hierarchical generative process to describe the data that are expressed as unit vectors (i.e., elements on the hypersphere). The task of posterior inference is to identify a set of latent topics, which are also unit vectors. This process is highly challenging due to a non-conjugate structure and the strict manifold constraints. None of the existing MCMC methods is both applicable to the task and scalable. We demonstrate that our methods are the most efficient methods to learn SAM on large datasets, with a good performance on testing data perplexity.

## 2 Preliminaries

We briefly review the basics of SG-MCMCs. Consider a Bayesian model with latent variable $q$, prior $\pi_0(q)$ and likelihood $\pi(x|q)$. Given a dataset $\mathcal{D} = \{x_d\}_{d=1}^D$, sampling from the posterior $\pi(q|\mathcal{D})$ by D-MCMCs requires computing the gradient of potential energy $\nabla U(q) \triangleq -\nabla \log \pi(q|\mathcal{D}) = -\nabla \log \pi_0(q) - \sum_{d=1}^D \nabla \log \pi(x_d|q)$, which is linear to data size $D$ thus not scalable. SG-MCMCs address this challenge by randomly drawing a subset $\mathcal{S}$ of $\mathcal{D}$ to build the stochastic gradient $\nabla_q \tilde{U}(q) \triangleq -\nabla_q \log \pi_0(q) - \frac{D}{|\mathcal{S}|} \sum_{x \in \mathcal{S}} \nabla_q \log \pi(x|q)$, a noisy but unbiased estimate.Under the i.i.d. assumption of $\mathcal{D}$, the central limit theorem holds: in the sense of convergence in distribution for large $D$,

$$\nabla_q \tilde{U}(q) = \nabla_q U(q) + \mathcal{N}(0, V(q)), \tag{1}$$

where we use $\mathcal{N}(\cdot, \cdot)$ to denote a Gaussian random variable and $V(q)$ is some covariance matrix.

The gradient noise raises challenging restrictions to the SG-MCMC dynamics. Ma et al. [19] then provide a recipe to construct correct dynamics. It claims that for a random variable $z$, given a Hamiltonian $H(z)$, a skew-symmetric matrix (curl matrix) $Q(z)$ and a positive definite matrix (diffusion matrix) $D(z)$, the dynamics defined by the following stochastic differential equation (SDE)

$$\mathrm{d}z = f(z)\mathrm{d}t + \sqrt{2D(z)}\mathrm{d}W(t) \tag{2}$$

has the unique stationary distribution $\pi(z) \propto \exp\{-H(z)\}$, where $W(t)$ is the Wiener process and

$$f(z) = -\left[D(z) + Q(z)\right]\nabla_z H(z) + \Gamma(z), \quad \Gamma_i(z) = \sum_j \frac{\partial}{\partial z_j}\left(D_{ij}(z) + Q_{ij}(z)\right). \tag{3}$$

The above dynamics is compatible with stochastic gradient. For SG-MCMCs, $z$ is usually an augmentation of the target variable $q$, and the Hamiltonian usually follows the form $H(z) = T(z) + U(q)$. Referring to Eqn. (1), $\nabla_q \tilde{H}(z) = \nabla_q H(z) + \mathcal{N}(0, V(q))$ and $\tilde{f}(z) = f(z) + \mathcal{N}(0, B(z))$, where $B(z)$ is the covariance matrix of the Gaussian noise passed from $\nabla_z \tilde{H}(z)$ to $\tilde{f}(z)$ through Eqn. (3). We informally rewrite $\mathrm{d}W(t)$ as $\mathcal{N}(0, \mathrm{d}t)$ and express dynamics Eqn. (2) as

$$\begin{aligned}
\mathrm{d}z &= f(z)\mathrm{d}t + \mathcal{N}(0, 2D(z)\mathrm{d}t) = f(z)\mathrm{d}t + \mathcal{N}(0, B(z)\mathrm{d}t^2) + \mathcal{N}\left(0, 2D(z)\mathrm{d}t - B(z)\mathrm{d}t^2\right) \\
&= \tilde{f}(z)\mathrm{d}t + \mathcal{N}\left(0, 2D(z)\mathrm{d}t - B(z)\mathrm{d}t^2\right).
\end{aligned} \tag{4}$$

This tells us that the same dynamics can be exactly expressed by stochastic gradient. Moreover, the recipe is complete: for any continuous Markov process defined by Eqn. (2) with a unique stationary distribution $\pi(z) \propto \exp\{-H(z)\}$, there exists a skew-symmetric matrix $Q(z)$ so that Eqn. (3) holds.

## 3 Stochastic Gradient Geodesic MCMC Methods

We now formally develop our SGGMC and gSGNHT. We will describe the task settings, develop the dynamics, and show how to simulate by 2nd-order integrators and stochastic gradient.

### 3.1 Technical Descriptions of the Settings

We first describe a Riemann manifold. Main concepts are depicted in Fig. 1. Let $\mathcal{M}$ be an $m$-dim Riemann manifold, which is covered by a set of local coordinate systems. Denote one of them by $(\mathcal{N}, \Phi)$, where $\mathcal{N} \subseteq \mathcal{M}$ is an open subset, and $\Phi : \mathcal{N} \to \Omega, Q \mapsto q$ with $\Omega \triangleq \Phi(\mathcal{N}) \subseteq \mathbb{R}^m$, $Q \in \mathcal{N}$ and $q \in \Omega$ is a homeomorphism. Additionally, transition mappings between any two intersecting local coordinate systems are required to be smooth. Denote the Riemann metric tensor under $(\mathcal{N}, \Phi)$ by $G(q)$, an $m \times m$ symmetric

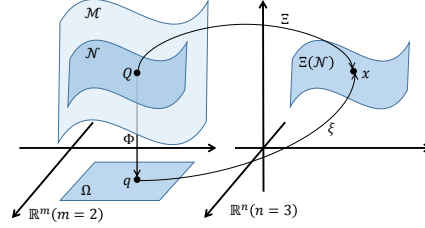

Figure 1: An illustration of manifold $\mathcal{M}$ with local coordinate system $(\mathcal{N}, \Phi)$ and embedding $\Xi$. See text for details.

positive-definite matrix. Another way to describe $\mathcal{M}$ is through *embedding* — a diffeomorphism $\Xi : \mathcal{M} \to \Xi(\mathcal{M}) \subseteq \mathbb{R}^n$ ($n \geq m$). In $(\mathcal{N}, \Phi)$, $\Xi$ can be embodied by a more sensible mapping $\xi \triangleq \Xi \circ \Phi^{-1} : \mathbb{R}^m \to \mathbb{R}^n, q \mapsto x$, which links the coordinate space and the embedded space. For convenience, we only consider *isometric* embeddings (whose existence is guaranteed [21]): $\Xi$ such that $G(q)_{ij} = \sum_{l=1}^n \frac{\partial \xi_l(q)}{\partial q_i} \frac{\partial \xi_l(q)}{\partial q_j}, 1 \leq i, j \leq m$ holds for any local coordinate system. Common manifolds are subsets of some $\mathbb{R}^n$, in which case the identity mapping (as $\Xi$) from $\mathbb{R}^n$ (where $\mathcal{M}$ is defined) to $\mathbb{R}^n$ (the embedded space) is isometric.

To define a distribution on a Riemann manifold, from which we want to sample, we need a measure. In the coordinate space $\mathbb{R}^m$, $\Omega$ naturally possesses the Lebesgue measure $\lambda^m(\mathrm{d}q)$, and the probability density can be defined in $\Omega$, which we denote as $\pi(q)$. In the embedded space $\mathbb{R}^n$, $\Xi(\mathcal{N})$ naturally possesses the Hausdorff measure $\mathcal{H}^m(\mathrm{d}x)$, and we denote the probability density w.r.t this measure as $\pi_\mathcal{H}(x)$. The relation between them can be found by $\pi_\mathcal{H}(\xi(q)) = \pi(q)/\sqrt{|G(q)|}$.

### 3.2 The Dynamics

We now construct our dynamics using the recipe [19] so that our dynamics naturally have the desired stationary distribution, leading to correct samples. It is important to note that the recipe only suits for dynamics in a Euclidean space. So we can only develop the dynamics in the coordinate space but not in the embedded space $\Xi(\mathcal{M})$, which is generally not Euclidean. However it is advantageous to simulate the dynamics in the embedded space (See Sec. 3.3).

**Dynamics for SGGMC**   Define the momentum in the coordinate space $p \in \mathbb{R}^m$ and the augmented variable $z = (q, p) \in \mathbb{R}^{2m}$. Define the Hamiltonian [2] $H(z) = U(q) + \frac{1}{2}\log|G(q)| + \frac{1}{2}p^\top G(q)^{-1}p$,

where $U(q) \triangleq -\log \pi(q)$. We define the Hamiltonian so that the canonical distribution $\pi(z) \propto \exp\{-H(z)\}$ marginalized w.r.t $p$ recovers the target distribution $\pi(q)$. For a symmetric positive definite $n \times n$ matrix $C$, define the diffusion matrix $D(z)$ and the curl matrix $Q(z)$ as

$$D(z) = \begin{pmatrix} 0 & 0 \\ 0 & M(q)^\top C M(q) \end{pmatrix}, \; Q(z) = \begin{pmatrix} 0 & -I \\ I & 0 \end{pmatrix},$$

where we define $M(q)_{n \times m} : M(q)_{ij} = \partial \xi_i(q)/\partial q_j$. So from Eqn. (2, 3), the dynamics

$$\begin{cases} dq = G^{-1}p\,dt \\ dp = -\nabla_q U dt - \dfrac{1}{2}\nabla_q \log|G|dt - M^\top C M G^{-1}p\,dt - \dfrac{1}{2}\nabla_q\big[p^\top G^{-1}p\big]dt + \mathcal{N}(0, 2M^\top C M dt) \end{cases}$$

(5)

has a unique stationary distribution $\pi(z) \propto \exp\{-H(z)\}$.

**Dynamics for gSGNHT**  Define $z = (q, p, \xi) \in \mathbb{R}^{2m+1}$, where $\xi \in \mathbb{R}$ is the thermostats. For a positive $C \in \mathbb{R}$, define the Hamiltonian $H(z) = U(q) + \frac{1}{2}\log|G(q)| + \frac{1}{2}p^\top G(q)^{-1}p + \frac{m}{2}(\xi - C)^2$, whose marginalized canonical distribution is $\pi(q)$ as desired. Define $D(z)$ and $Q(z)$ as

$$D(z) = \begin{pmatrix} 0 & 0 & 0 \\ 0 & CG(q) & 0 \\ 0 & 0 & 0 \end{pmatrix}, \; Q(z) = \begin{pmatrix} 0 & -I & 0 \\ I & 0 & p/m \\ 0 & -p^\top/m & 0 \end{pmatrix},$$

Then by Eqn. (2, 3) the proper dynamics of gSGNHT is

$$\begin{cases} dq = G^{-1}p\,dt \\ dp = -\nabla_q U dt - \dfrac{1}{2}\nabla_q \log|G|dt - \xi p\,dt - \dfrac{1}{2}\nabla_q\big[p^\top G^{-1}p\big]dt + \mathcal{N}(0, 2CG dt) \\ d\xi = (\dfrac{1}{m}p^\top G^{-1}p - 1)dt \end{cases}$$

(6)

These two dynamics are novel. They are extensions of the dynamics of SGHMC and SGNHT to Riemann manifolds, respectively. Conceiving the dynamics in this form is also intended for the convenience to develop 2nd-order geodesic integrators, which differs from SGRHMC.

### 3.3  Simulation with 2nd-order Geodesic Integrators

In this part we develop our integrators by following the *symmetric splitting integrator* (SSI) scheme [8], which is guaranteed to be of 2nd-order. The idea of SSI is to first split the dynamics into parts with each analytically solvable, then alternately simulate each exactly with the analytic solutions. Although also SSI, the integrator of GMC does not fit our dynamics where diffusion arises. But we adopt its embedding technique to get rid of any local coordinate system thus release the global coordinate system requirement. So we will solve and simulate the split dynamics in the isometrically embedded space, where everything is expressed by the position $x = \xi(q)$ and the velocity $v = \dot{x}$ (which is actually the momentum in the isometrically embedded space, see Appendix C; the overhead dot means time derivative), instead of $q$ and $p$.

**Integrator for SGGMC**  We first split dynamics (5) into sub-SDEs with each analytically solvable:

$$A:\begin{cases} dq = G^{-1}p\,dt \\ dp = -\dfrac{1}{2}\nabla_q[p^\top G^{-1}p]dt \end{cases}, B:\begin{cases} dq = 0 \\ dp = -M^\top C M G^{-1}p\,dt \end{cases}, O:\begin{cases} dq = 0 \\ dp = -\nabla_q U(q)dt - \dfrac{1}{2}\nabla_q\log|G(q)|dt \\ \quad\quad + \mathcal{N}(0, 2M^\top C M dt) \end{cases}.$$

As noted in GMC, the solution of dynamics $A$ is the geodesic flow of the manifold [1]. Intuitively, dynamics $A$ describes motion with no force so a particle moves freely on the manifold, e.g. the *uniform motion* in Euclidean space, and motion along *great circles* (velocity rotating with varying tangents along the trajectory) on hypersphere $\mathbb{S}^{d-1} \triangleq \{x \in \mathbb{R}^d | \|x\| = 1\}$ ($\|\cdot\|$ denotes $\ell_2$-norm). The evolution of the position and velocity of this kind is the geodesic flow. We require an explicit form of the geodesic flow in the embedded space. For $\mathbb{S}^{d-1}$,

$$\begin{cases} x(t) = x(0)\cos(\alpha t) + \big(v(0)/\alpha\big)\sin(\alpha t) \\ v(t) = -\alpha x(0)\sin(\alpha t) + v(0)\cos(\alpha t) \end{cases}$$

(7)

is the geodesic flow expressed by the embedded variables $x$ and $v$, where $\alpha = \|v(0)\|$.

By details in [7] or Appendix A, dynamics $B$ and $O$ are solved as

$$B:\begin{cases} x(t)=x(0) \\ v(t)=\mathrm{expm}\{-\Lambda\big(x(0)\big)Ct\}v(0) \end{cases}, O:\begin{cases} x(t)=x(0) \\ v(t)=v(0)+\Lambda\big(x(0)\big)\big[-\nabla_x U_{\mathcal{H}}\big(x(0)\big)t+\mathcal{N}(0,2Ct)\big] \end{cases},$$

where $U_{\mathcal{H}}(x) \triangleq -\log \pi_{\mathcal{H}}(x)$, $\mathrm{expm}\{\cdot\}$ is the matrix exponent, and $\Lambda(x)$ is the projection onto the tangent space at $x$ in the embedded manifold. For $\mathbb{R}^n$, $\Lambda(x) = I_n$ (the identity mapping in $\mathbb{R}^n$) and for $\mathbb{S}^{n-1}$ embedded in $\mathbb{R}^n$, $\Lambda(x) = I_n - xx^\top$ (see Appendix A.3).

We further reduce dynamics $B$ for *scalar C*: $v(t) = \Lambda(x(0)) \exp\{-Ct\}v(0) = \exp\{-Ct\}v(0)$, by noting that $\exp\{-Ct\}$ is a scalar and $v(0)$ already lies on the tangent space at $x(0)$. To illustrate this form, we expand the exponent for small $t$ and get $v(t) = (1 - Ct)v(0)$, which is exactly the action of a friction dissipating energy to control injected noise, as proposed in SGHMC. Our investigation reveals that this form holds generally for $v$ as the momentum in the isometrically embedded space, but not the usual momentum $p$ in the coordinate space. In SGHMC, $v$ and $p$ are undistinguishable, but in our case $v$ can only lie in the tangent space and $p$ is arbitrary in $\mathbb{R}^m$.

**Integrator for gSGNHT**  We split dynamics (6) in a similar way:

$$A:\begin{cases} \mathrm{d}q =G^{-1}p\mathrm{d}t \\ \mathrm{d}p =-\frac{1}{2}\nabla_q\big[p^\top G^{-1}p\big]\mathrm{d}t \\ \mathrm{d}\xi =\Big(\frac{1}{m}p^\top G^{-1}p-1\Big)\mathrm{d}t \end{cases}, B:\begin{cases} \mathrm{d}q =0 \\ \mathrm{d}p =-\xi p\,\mathrm{d}t \\ \mathrm{d}\xi =0 \end{cases}, O:\begin{cases} \mathrm{d}q =0 \\ \mathrm{d}p =-\nabla_q U\mathrm{d}t-\frac{1}{2}\nabla_q \log|G|\,\mathrm{d}t \\ \qquad +\mathcal{N}(0,2CG\mathrm{d}t) \\ \mathrm{d}\xi =0 \end{cases}.$$

For dynamics $A$, the solution of $q$ and $p$ is again the geodesic flow. To solve $\xi$, we first figure out that for dynamics $A$, $p^\top G^{-1}p$ is constant: $\frac{\mathrm{d}}{\mathrm{d}t}\big[p^\top G(q)^{-1}p\big] = \nabla_q\big[p^\top G(q)^{-1}p\big]^\top \dot{q}+2\big[G(q)^{-1}p\big]^\top \dot{p} = -2\dot{p}^\top \dot{q} + 2\dot{q}^\top \dot{p} = 0$. Alternatively we note that $\frac{1}{2}p^\top G^{-1}p = \frac{1}{2}v^\top v$ is the kinetic energy [3] conserved by motion with no force. Now the evolution of $\xi$ can be solved as $\xi(t) = \xi(0)+\big(\frac{1}{m}v(0)^\top v(0)-1\big)t$.

Dynamics $O$ is identical to the one of SGGMC. Dynamics $B$ can be solved similarly with only $v$ updated: $v(t) = \exp\{-\xi(0)t\}v(0)$. Expansion of this recovers the dissipation of energy by an adaptive friction proposed by SGNHT, and we extend it to an embedded space.

Now we consider incorporating stochastic gradient. Only the common dynamics $O$ is affected. Similar to Eqn. (1), we express the stochastic gradient as $\nabla_x \tilde{U}_{\mathcal{H}}(x) = \nabla_x U_{\mathcal{H}}(x) + \mathcal{N}(0, V(x))$, then reformulate the solution of dynamics $O$ as

$$v(t) = v(0) + \Lambda\big(x(0)\big) \cdot \Big[-\nabla_x\tilde{U}_{\mathcal{H}}\big(x(0)\big)t + \mathcal{N}\Big(0, 2Ct-V\big(x(0)\big)t^2\Big)\Big]. \tag{8}$$

To estimate the usually unknown $V(x)$, a simple way is just to take it as zero, in the sense that $V(x)t^2$ is a higher order infinitesimal of $2Ct$ for $t$ as a small simulation step size. Another way to estimate $V(x)$ is by the empirical Fisher information, as is done in [2].

Finally, as SSI suggests, we simulate the complete dynamics by exactly simulating these solutions alternately in an "ABOBA" pattern. For a time step size of $\varepsilon$, dynamics $A$ and $B$ advance by $\varepsilon/2$ for once and dynamics $O$ by $\varepsilon$. As other SG-MCMCs, we omit the unscalable Metropolis-Hastings test. But the consistency is still guaranteed [8] of e.g. the estimation by averaging over samples drawn from SG-MCMCs. Algorithms of SGGMC and gSGNHT are listed in Appendix E.

## 4  Application to Spherical Admixture Model

We now apply SGGMC/gSGNHT to solve the challenging task of posterior inference in Spherical Admixture Model (SAM) [24]. SAM is a Bayesian topic model for spherical data (each datum is in some $\mathbb{S}^{d-1}$), such as the *tf-idf* representation of text data. It enables more feature representations for hierarchical Bayesian models, and have the benefit over Latent Dirichlet Allocation (LDA) [5] to directly model the absence of words. The structure of SAM is shown in Fig. 2. Each document $v_d$, each topic $\beta_k$, the corpus mean $\mu$ and the hyper-parameter $m$ are all in $\mathbb{S}^{V-1}$ with $V$ the vocabulary size. Each topic proportion $\theta_d$ is in $(K-1)$-dim simplex with $K$ the number of topics.

SAM uses the von Mises-Fisher distribution (vMF) (see e.g. [20]) to model variables on hyperspheres. The vMF on $\mathbb{S}^{d-1}$ with mean $\mu \in \mathbb{S}^{d-1}$ and concentration parameter $\kappa \in \mathbb{R}^+$ has *pdf* (w.r.t the Hausdorff measure) $\text{vMF}(x|\mu,\kappa) = c_d(\kappa)\exp\{\kappa\mu^\top x\}$, where $c_d(\kappa) = \kappa^{d/2-1}/\big((2\pi)^{d/2}I_{d/2-1}(\kappa)\big)$ and $I_r(\cdot)$ denotes the modified Bessel function of the first kind and order $r$. Then the generating process of SAM is

- Draw $\mu \sim \text{vMF}(\mu|m,\kappa_0)$;
- For $k = 1,\ldots,K$, draw topic $\beta_k \sim \text{vMF}(\beta_k|\mu,\sigma)$;
- For $d = 1,\ldots,D$, draw $\theta_d \sim \text{Dir}(\theta_d|\alpha)$ and $v_d \sim \text{vMF}(v_d|\bar{v}(\beta,\theta_d),\kappa)$,

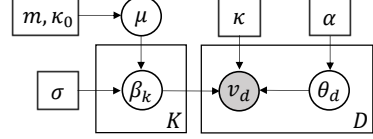

where $\bar{v}(\beta,\theta_d) \triangleq \frac{\beta\theta_d}{\|\beta\theta_d\|}$ with $\beta \triangleq (\beta_1,\ldots,\beta_K)$ is an approximate spherical weighted mean of topics. The joint distribution of $\big(v \triangleq (v_1,\ldots,v_D),\beta,\theta \triangleq (\theta_1,\ldots,\theta_K),\mu\big)$ can be known.

Figure 2: An illustration of SAM model structure.

The inference task is to estimate the topic posterior $\pi(\beta|v)$. As it is intractable, [24] provides a mean-field variational inference method and solves an optimization problem under spherical constraint, which is tackled by repeatedly normalizing. However, this treatment is not applicable to most sampling methods since it may corrupt the distribution of the samples. [24] tries a simple adaptive Metropolis-Hastings sampler with undesirable results, and no more attempt of sampling methods appears. Due to the deficiency of global coordinate system of hypersphere, most Riemann manifold samplers including SGRLD and SGRHMC fail. To our knowledge, only CHMC and GMC are suitable, yet not scalable. Our samplers are appropriate for the task, with the advantage of scalability.

Now we present our inference method that uses SGGMC/gSGNHT to directly sample from $\pi(\beta|v)$. First we note that $\mu$ can be collapsed analytically and the marginalized distribution of $(v,\beta,\theta)$ is:

$$\pi(v,\beta,\theta) = c_V(\kappa_0)c_V(\sigma)^K c_V(\|\bar{m}(\beta)\|)^{-1} \prod_{d=1}^{D} \text{Dir}(\theta_d|\alpha)\text{vMF}(v_d|\bar{v}(\beta,\theta_d),\kappa), \qquad (9)$$

where $\bar{m}(\beta) \triangleq \kappa_0 m + \sigma\sum_{k=1}^{K}\beta_k$. To sample from $\pi(\beta|v)$ using our samplers, we only need to know a stochastic estimate of the gradient of potential energy $\nabla_\beta U(\beta|v) \triangleq -\nabla_\beta\log\pi(\beta|v)$, which can be estimated by adopting the technique used in [11]: $\nabla_\beta\log\pi(\beta|v) =$

$$\frac{1}{\pi(\beta|v)}\nabla_\beta\int\pi(\beta,\theta|v)\mathrm{d}\theta = \int\frac{\pi(\beta,\theta|v)}{\pi(\beta|v)}\frac{\nabla_\beta\pi(\beta,\theta|v)}{\pi(\beta,\theta|v)}\mathrm{d}\theta = \mathbb{E}_{\pi(\theta|\beta,v)}\left[\nabla_\beta\log\pi(\beta,\theta|v)\right],$$

where $\nabla_\beta\log\pi(\beta,\theta|v) = \nabla_\beta\log\pi(v,\beta,\theta)$ is known, and the expectation can be estimated by averaging over a set of samples $\{\theta^{(n)}\}_{n=1}^N$ from $\pi(\theta|v,\beta)$: $\nabla_\beta U(\beta|v) \approx \frac{1}{N}\sum_{n=1}^N\nabla_\beta\log\pi(v,\beta,\theta^{(n)})$. To draw $\{\theta^{(n)}\}_{n=1}^N$, noting the simplex constraint and that the target distribution $\pi(\theta|v,\beta)$ is known up to a constant multiplier, we use GMC to do the task.

To scale up, we use a subset $\{d(s)\}_{s=1}^S$ of indices of randomly chosen items from the whole data set to get a stochastic estimate for each $\nabla_\beta\log\pi(v,\beta,\theta^{(n)})$. The final stochastic gradient is:

$$\nabla_\beta\tilde{U}(\beta|v) \approx \nabla_\beta\log c_V(\|\bar{m}(\beta)\|) - \kappa\frac{D}{NS}\sum_{n=1}^N\sum_{s=1}^S v_{d(s)}^\top\bar{v}(\beta,\theta_{d(s)}^{(n)}). \qquad (10)$$

The inference algorithm for SAM by SGGMC/gSGNHT is summarized in Alg. 3 in Appendix E.

## 5 Experiments

We present empirical results on both synthetic and real datasets to prove the accuracy and efficiency of our methods. All target densities are expressed in the embedded space w.r.t the Hausdorff measure so we omit the subscript "$\mathcal{H}$". Synthetic experiments are only for SGGMC since the advantage to use thermostats has been shown by [10] and the effectiveness of gSGNHT is presented on real datasets. Detailed settings of the experiments are provided in Appendix F.

### 5.1 Toy Experiment

We first present the utility and check the correctness of SGGMC by a greenhouse experiment with known stochastic gradient noise. Consider sampling from a circle ($\mathbb{S}^1$) for easy visualization. We set the target distribution such that the potential energy is $U(x) = -\log\big(\exp\{5\mu_1^\top x\} + 2\exp\{5\mu_2^\top x\}\big)$, where $x,\mu_1,\mu_2 \in \mathbb{S}^1$ and $\mu_1 = -\mu_2 = \frac{\pi}{3}$ (angle from $+x$ direction). The stochastic gradient is

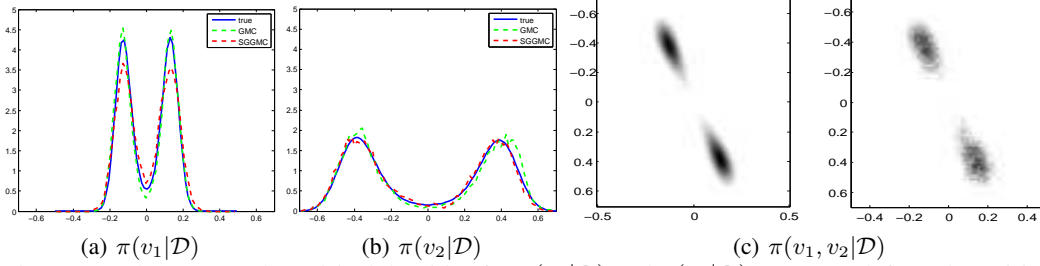

(a) $\pi(v_1|\mathcal{D})$       (b) $\pi(v_2|\mathcal{D})$       (c) $\pi(v_1, v_2|\mathcal{D})$

Figure 4: (a-b): True and empirical densities for $\pi(v_1|\mathcal{D})$ and $\pi(v_2|\mathcal{D})$. (c) True (left) and empirical by SGGMC (right) densities for $\pi(v_1, v_2|\mathcal{D})$.

produced by corrupting with $\mathcal{N}(0, 1000I)$, whose variance is used as $V(x)$ in Eqn. (8) for sampling. Fig. 3(a) shows 100 samples from SGGMC and empirical distribution of 10,000 samples in the embedded space $\mathbb{R}^2$. True and empirical distributions are compared in Fig. 3(b) in angle space (local coordinate space). We see no obvious corruption of the result when using stochastic gradient.

It should be stressed that although it is possible to apply scalable methods like SGRLD in spherical coordinate systems (almost global ones), it is too troublesome to work out the form of e.g. Riemann metric tensor, and special treatments like reflection at boundaries have to be considered. Numerical instability at boundaries also tends to appear. All these will get even worse in higher dimensions. Our methods work in embedded spaces, so all these issues are bypassed and can be elegantly extended to high dimensions.

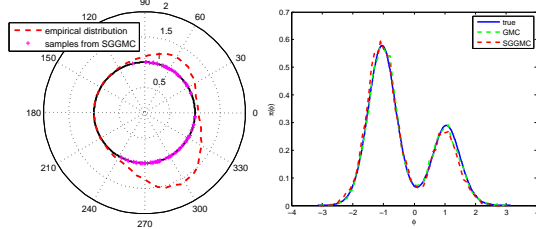

(a) samples by SGGMC in the embedded space    (b) distribution comparison in angle space

Figure 3: Toy experiment results: (a) samples and empirical distribution of SGGMC; (b) comparison of true and empirical distributions.

## 5.2 Synthetic Experiment

We then test SGGMC on a simple Bayesian posterior estimation task. We adopt a model with similar structure as the one used in [29]. Consider a mixture model of two vMFs on $\mathbb{S}^1$ with equal weights:

$$\pi(v_1) = \text{vMF}(v_1|e_1, \kappa_1), \ \pi(v_2) = \text{vMF}(v_2|e_1, \kappa_2), \ \pi(x_i|v_1, v_2) \propto \text{vMF}(x_i|v_1, \kappa_x) + \text{vMF}(x_i|\mu, \kappa_x),$$

where $e_1 = (1, 0)$ and $\mu \triangleq (v_1 + v_2)/\|v_1 + v_2\|$. The task is to infer the posterior $\pi(v_1, v_2|\mathcal{D})$, where $\mathcal{D} = \{x_i\}_{i=1}^{D=100}$ is our synthetic data that is generated from the likelihood with $v_1 = -\frac{\pi}{24}, v_2 = \frac{\pi}{8}$ and $\kappa_1 = \kappa_2 = \kappa_x = 20$ by GMC. SGGMC uses empirical Fisher information in the way of [2] for $V(x)$ in Eqn. (8), and uses 10 for batch size. Fig. 4(a-b) show the true and empirical marginal posteriors of $v_1$ and $v_2$, and Fig. 4(c) presents empirical joint posterior by samples from SGGMC and its true density. We see that samples from SGGMC exhibit no observable corruption when a mini-batch is used, and fully explore the two modes and the strong correlation of $v_1$ and $v_2$. [4]

## 5.3 Spherical Admixture Models

**Setups** For baselines, we compare with the mean-field variational inference (**VI**) by [24] and its stochastic version (**StoVI**) based on [15], as well as GMC methods. It is problematic for GMC to directly sample from the target distribution $\pi(\beta|v)$ since the potential energy is hard to estimate, which is required for Metropolis-Hastings (MH) test in GMC. An approximate Monte Carlo estimation is provided in Appendix B and the corresponding method for SAM is **GMC-apprMH**. An alternative is **GMC-bGibbs**, which adopts blockwise Gibbs sampling to alternately sample from $\pi(\beta|\theta, v)$ and $\pi(\theta|\beta, v)$ (both known up to a constant multiplier) using GMC.

We evaluate the methods by *log-perplexity* — the average of negative log-likelihood on a held-out test set $\mathcal{D}_{test}$. Variational methods produce a single point estimate $\hat{\beta}$ and the log-perplexity is log-perp $= -\frac{1}{|\mathcal{D}_{test}|} \sum_{d \in \mathcal{D}_{test}} \log \pi(v_d|\hat{\beta})$. Sampling methods draw a set of samples $\{\beta^{(m)}\}_{m=1}^M$ and log-perp $= -\frac{1}{|\mathcal{D}_{test}|} \sum_{d \in \mathcal{D}_{test}} \log(\frac{1}{M} \sum_{m=1}^M \pi(v_d|\beta^{(m)}))$. In both cases the intractable $\pi(v_d|\beta)$ needs to be estimated. By noting that $\pi(v_d|\beta) = \int \pi(v_d, \theta_d|\beta) d\theta_d = \mathbb{E}_{\pi(\theta_d|\beta)}[\pi(v_d|\beta, \theta_d)]$, we

estimate it by averaging $\pi(v_d|\beta, \theta_d^{(n)})$ (exactly known from the generating process) over samples $\{\theta_d^{(n)}\}_{n=1}^{N}$ drawn from $\pi(\theta_d|\beta) = \pi(\theta_d) = \mathrm{Dir}(\alpha)$, the prior of $\theta_d$. The log-perplexity is not comparable among different models so we exclude LDA from our baseline.

We show the performance of all methods on a small and a large dataset. Hyper-parameters of SAM are fixed while training and set the same for all methods. $V(x)$ in Eqn. (8) is taken zero for **SGGMC/gSGNHT**. All sampling methods are implemented [5] in C++ and *fairly* parallelized by OpenMP. **VI/StoVI** are run in MATLAB codes by [24] and we only use their final scores for comparison. Appendix F gives further implementation details, including techniques to avoid overflow.

**On the small dataset**
The small dataset is the
20News-different dataset
used by [24], which con-
sists of 3 categories from
20Newsgroups dataset. It
is small (1,666 training and
1,107 test documents) so
we have the chance to see
the eventual results of all
methods. We use 20 topics
and 50 as the batch size.

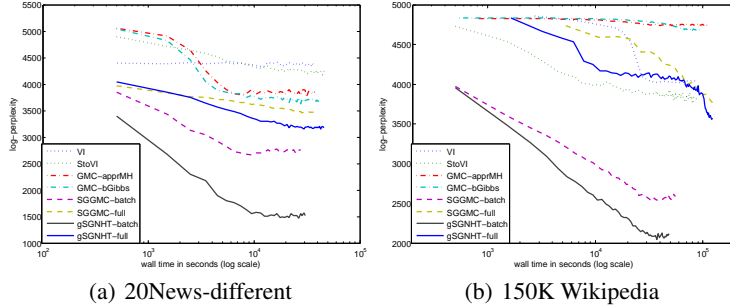

(a) 20News-different         (b) 150K Wikipedia

Figure 5: Evolution of log-perplexity along wall time of all methods
on (a) 20News-different dataset and (b) 150K Wikipedia subset.

Fig. 5(a) shows the perfor-
mance of all methods. We
can see that our **SGGMC** and **gSGNHT** perform better than others. **VI** converges swiftly but cannot go any lower due to the intrinsic gap between the mean-field variational distribution and the true posterior. **StoVI** converges slower than **VI** in this small scale case, and exhibits the same limit. All sampling methods eventually go below variational methods, and ours go the lowest. **gSGNHT** shows its benefit to outperform **SGGMC** under the same setting. For our methods, an appropriately smaller batch size achieves a better result due to the speed-up by subsampling. Note that even the full-batch **SGGMC** and **gSGNHT** outperform GMC variants. This may be due to the randomness in the dynamics helps jumping out of one local mode to another for a better exploration.

**On the large dataset**   For the large dataset, we use a subset of the Wikipedia dataset with 150K training and 1K test documents, to challenge the scalability of all the methods. We use 50 topics and 100 as the batch size. Fig. 5(b) shows the outcome. We see that the gap between our methods and other baselines gets larger, indicating our scalability. Bounded curves of **VI/StoVI**, the advantage of using thermostats and subsampling speed-up appear again. Our full-batch versions are still better than GMC variants. **GMC-apprMH** and **GMC-bGibbs** scale badly; they converge slowly in this case.

## 6   Conclusions and Discussions

We propose SGGMC and gSGNHT, SG-MCMCs for scalable sampling from manifolds with known geodesic flow. They are saliently efficient on their applications. Novel dynamics are constructed and 2nd-order geodesic integrators are developed. We apply the methods to SAM topic model for more accurate and scalable inference. Synthetic experiments verify the validity and experiments for SAM on real-world data shows an obvious advantage in accuracy over variational inference methods and in scalability over other applicable sampling methods. There remains possible broad applications of our methods, including models involving vMF (e.g. mixture of vMF [4, 14, 28], DP mixture of vMF [12, 3, 27]), constraint distributions [17] (e.g. truncated Gaussian), and distributions on Stiefel manifold (e.g. Bayesian matrix completion [25]), where the ability of scale-up will be appealing.

**Acknowledgments**

The work was supported by the National Basic Research Program (973 Program) of China (No. 2013CB329403), National NSF of China Projects (Nos. 61620106010, 61322308, 61332007), the Youth Top-notch Talent Support Program, and Tsinghua Initiative Scientific Research Program (No. 20141080934).

## Footnotes

[2]Another derivation of the momentum and the Hamiltonian originated from physics in both coordinate and embedded spaces is provided in Appendix C.

[3] $p^\top G^{-1}p = (G^{-1}p)^\top G(G^{-1}p) = \dot{q}^\top (M^\top M)\dot{q} = (M\dot{q})^\top (M\dot{q}) = v^\top v$ for an isometric embedding.

[4]Appendix D provides a rationale on the shape of the joint posterior.

[5]All the codes and data can be found at `http://ml.cs.tsinghua.edu.cn/~changliu/sggmcmc-sam/`.

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
