[Supplementary Material]

# Supplementary Material for:
# Stochastic Gradient Geodesic MCMC Methods

**Chang Liu[†], Jun Zhu[†], Yang Song[‡*]**
[†] Dept. of Comp. Sci. & Tech., TNList Lab; Center for Bio-Inspired Computing Research
[†] State Key Lab for Intell. Tech. & Systems, Tsinghua University, Beijing, China
[‡] Dept. of Physics, Tsinghua University, Beijing, China
{chang-li14@mails, dcszj@}.tsinghua.edu.cn; songyang@stanford.edu

## Appendix A: Details for Solving Dynamics $B$ and $O$ in the Embedded Space

In this section, we present details used in Sec. 3.3 of the main paper to solve dynamics $B$ and $O$ in the embedded space for releasing local coordinate systems and a cheap simulation rule. We only show the details for SGGMC, and the procedure is quite similar for gSGNHT. The technique is adapted from [2], and we reformulate it here for completeness and easier comprehension.

### A.1: Main Details

Dynamics $B$ and $O$ of SGGMC in the coordinate space (see Sec. 3.3.1 of the main paper) are:

$$B : \begin{cases} \mathrm{d}q = 0 \\ \mathrm{d}p = -M^\top C M G^{-1} p\, \mathrm{d}t \end{cases}, O : \begin{cases} \mathrm{d}q = 0 \\ \mathrm{d}p = -\nabla_q U(q)\, \mathrm{d}t - \dfrac{1}{2}\nabla_q \log|G(q)|\, \mathrm{d}t + \mathcal{N}(0, 2M^\top C M \mathrm{d}t) \end{cases} .$$

We do the transformation into the embedded space based on the map $x = \xi(q)$. Note that for both dynamics, $q$ is constant so $\dot{x} = (\nabla_q \xi)^\top \dot{q} = 0$. From dynamics $A$ in Sec. 3.3.1 (or the definition of momentum in Appendix C), we have $p = G(q)\dot{q}$ and by some calculus we have $\nabla_q = M^\top \nabla_x$. From Sec. 3.1 we have $\pi_{\mathcal{H}}(x) = \pi(q)/\sqrt{|G(q)|}$ and $U_{\mathcal{H}}(x) \triangleq -\log \pi_{\mathcal{H}}(x) = U(q) + \frac{1}{2}\log|G(q)|$. So we have

$$B : \begin{cases} \mathrm{d}x = 0 \\ G(q)\mathrm{d}\dot{q} = -M^\top C M\dot{q}\, \mathrm{d}t \end{cases}, O : \begin{cases} \mathrm{d}x = 0 \\ G(q)\mathrm{d}\dot{q} = -M^\top \nabla_x U_{\mathcal{H}}(x)\mathrm{d}t + M^\top \mathcal{N}(0, 2C\mathrm{d}t) \end{cases} .$$

We then left multiply $M(q)G(q)^{-1}$ for both momentum equations and have

$$B : \begin{cases} \mathrm{d}x = 0 \\ \mathrm{d}(M\dot{q}) = -MG(q)^{-1}M^\top C M\dot{q}\, \mathrm{d}t \end{cases}, O : \begin{cases} \mathrm{d}x = 0 \\ \mathrm{d}(M\dot{q}) = -MG(q)^{-1}M^\top \nabla_x U_{\mathcal{H}}(x)\mathrm{d}t \\ \qquad\qquad + MG(q)^{-1}M^\top \mathcal{N}(0, 2C\mathrm{d}t) \end{cases} .$$

By definition $v \triangleq \dot{x} = M\dot{q}$; from the isometric property of the Riemann metric $G(q)$, we have $G = M^\top M$. So finally we have

$$B : \begin{cases} \mathrm{d}x = 0 \\ \mathrm{d}v = -M(M^\top M)^{-1}M^\top C v\, \mathrm{d}t \end{cases}, O : \begin{cases} \mathrm{d}x = 0 \\ \mathrm{d}v = M(M^\top M)^{-1}M^\top \big(-\nabla_x U_{\mathcal{H}}(x)\mathrm{d}t + \mathcal{N}(0, 2C\mathrm{d}t)\big) \end{cases} .$$

We can write $M(x) = M(\xi^{-1}(x))$ so $M$ can be expressed in the embedded space. Now we complete the transformation into the embedded space.

---

[*]JZ is the corresponding author; YS is with Department of Computer Science, Stanford University, CA.

Note that the matrix $M(M^\top M)^{-1}M^\top$, which we denote as $\Lambda(x)$, is the orthogonal projection onto the column space of $M$ (see Appendix A.2), which is $T_x(\Xi(\mathcal{M}))$, the tangent space of the embedded manifold at $x$ (a conclusion from differential geometry). [2] proposes an alternative expression for $\Lambda(x)$ that can be more intuitively constructed and is computationally cheaper. Appendix A.2 shows the interpretation of the two expressions and Appendix A.3 derives $\Lambda(x)$ for hyperspheres based on both expressions. By noting that $x$ is constant for both dynamics, we can analytically solve them

$$B: \begin{cases} x(t) = x(0) \\ v(t) = \mathrm{expm}\left\{-\Lambda\big(x(0)\big)Ct\right\}v(0) \end{cases}, O: \begin{cases} x(t) = x(0) \\ v(t) = v(0) + \Lambda\big(x(0)\big)\big[-\nabla_x U_\mathcal{H}\big(x(0)\big)t + \mathcal{N}(0, 2Ct)\big] \end{cases}$$

as given in the main paper.

## A.2: Interpretation of the Two Expressions of the Orthogonal Projection $\Lambda(x)$

We first interpret that $\Lambda(x) \triangleq M(M^\top M)^{-1}M^\top$ is an orthogonal projection onto the column space $C(M)$ of $M$. In our case, $M_{n \times m}$ has full column rank (differential geometry conclusion) so $C(M)$ is an $m$-dim subspace (hyperplane) of $\mathbb{R}^n$. An orthogonal projection of $x \in \mathbb{R}^n$ onto $C(M)$ gives the following decomposition

$$x = y + z,$$

where $y \in C(M)$, $z \in (C(M))^\perp$ so $y^\top z = 0$. Express $y$ as $y = M\theta_x$, where $\theta \in \mathbb{R}^m$ acts as the coordinate of elements in $C(M)$. Then $y^\top z = 0$ gives $(M\theta_x)^\top(x - M\theta_x) = 0$, which is $\theta_x^\top(M^\top x - M^\top M\theta_x) = 0$, and we solve

$$\theta_x = (M^\top M)^{-1}M^\top x.$$

So $y = M(M^\top M)^{-1}M^\top x$, revealing the meaning of $\Lambda(x)$ as the orthogonal projection onto $C(M)$.

Now we introduce another expression for $\Lambda(x)$ proposed in [2]. Let $N_{n \times (n-m)}$ be a set of orthonormal basis (collected in columns) of $(C(M))^\perp$ so that $N^\top N = I_{n-m}$. Express $z$ as $z = N\phi_x$, where $\phi \in \mathbb{R}^{n-m}$ acts as the coordinate of elements in $(C(M))^\perp$. Then $y^\top z = 0$ gives $(N\phi_x)^\top(x - N\phi_x) = 0$, and we solve

$$\phi_x = (N^\top N)^{-1}N^\top x = N^\top x.$$

So $y = x - NN^\top x$, revealing the meaning of $I_n - NN^\top$ as the same projection. This expression is proposed because $N$ is usually more sensible, especially for hyperspheres as shown in Appendix A.3. It also saves computation since only one thin matrix multiplication is required, instead of more than three matrix multiplications and possibly a matrix inversion for the former expression.

## A.3: A Derivation of the Projection $\Lambda(x)$ for the Hypersphere Example

First we need to describe a hypersphere in depth. A $(d-1)$-dim hypersphere $\mathbb{S}^{d-1} \triangleq \{x \in \mathbb{R}^d \mid \|x\| = 1\}$ is defined as a subset of $\mathbb{R}^n$, so we can *isometrically* embed it into $\mathbb{R}^d$ by an identity mapping $\Xi: \mathbb{S}^{d-1} \to \mathbb{R}^d, x \mapsto x$. To get $M(q)$, we need to specify a local coordinate system for $\mathbb{S}^{d-1}$. We select $\mathcal{N} = \{x \in \mathbb{S}^{d-1} \mid x_d > 0\}$ i.e. the upper semi-hypersphere, and $\Omega = \{q = (q_1, \dots, q_{d-1})^\top \mid \sum_{i=1}^{d-1} q_i^2 < 1\} \subset \mathbb{R}^{d-1}$. For $x \in \mathcal{N}$, define $\Phi(x) = (x_1, \dots, x_{d-1})^\top \in \Omega$. Then $(\mathcal{N}, \Phi)$ is a local coordinate system for $\mathbb{S}^{d-1}$ and $\xi(q) = (q_1, \dots, q_{d-1}, \xi_d)^\top \in \mathbb{R}^d$, where $\xi_d = \sqrt{1 - \sum_{i=1}^{d-1} q_i^2}$. By definition, we have

$$M(q) = \begin{pmatrix} I_{d-1} \\ -q^\top/\xi_d \end{pmatrix}, G(q) = M^\top M = I_{d-1} + \frac{qq^\top}{\xi_d^2}.$$

By the Sherman-Morrison formula, we have

$$G(q)^{-1} = I_{d-1} - qq^\top.$$

So according to the first expression of $\Lambda(x)$, we have

$$\Lambda(\xi(q)) = MG^{-1}M^\top = \begin{pmatrix} I_{d-1} - qq^\top & -\xi_d q \\ -\xi_d q^\top & 1 - \xi_d^2 \end{pmatrix} = I_d - \xi(q)\xi(q)^\top,$$

or $\Lambda(x) = I_d - xx^\top, x \in \Xi(\mathcal{N})$. We successively select similar local coordinate systems (i.e. select $\Omega$ as lower semi-hypersphere, east semi-hypersphere, etc.) until $\mathbb{S}^{d-1}$ can be covered by these coordinates. For each of these local coordinate systems, we have the same conclusion. To sum up, we have $\Lambda(x) = I_d - xx^\top, x \in \Xi(\mathcal{M})$.

Another way to implement the projection is though the orthonormal basis $N(x)_{n \times (n-m)}$ of the orthogonal complement of the tangent space. For the hypersphere $\mathbb{S}^{d-1}$, the tangent space $T_x\mathbb{S}^{d-1}$ is intuitively a $(d-1)$-dim subplane in $\mathbb{R}^d$ perpendicular to the direction of radius, which is just $x$. The orthogonal complement of the plane is the line in the direction of $x$. Thus $N(x) = x$ and by the second expression $\Lambda(x) = I_d - xx^\top$, the same as derived by the first expression. We see that this derivation is more neat.

## Appendix B: Estimation of the Potential Energy $U(\beta|v)$ for SAM

For the SAM inference task, the potential energy $U(\beta|v) \triangleq -\log \pi(\beta|v)$ is required for GMC to directly sample from $\pi(\beta|v)$. It is involved since we do not know the closed form of $U(\beta|v)$, even ignoring a shifting constant. By referring to the unbiased Monte Carlo estimation of its gradient $-\nabla_\beta \log \pi(\beta|v) = -\mathbb{E}_{\pi(\theta|v,\beta)}[\nabla_\beta \log \pi(\beta, \theta|v)]$ as provided in Sec. 4 of the main paper, one possible estimation can be formed by samples $\{\theta^{(n)}\}_{n=1}^N$ from $\pi(\theta|v, \beta)$:

$$U(\beta|v) \approx -\frac{1}{N} \sum_{n=1}^N \log \pi(\beta, \theta^{(n)}|v) + \text{const.} \tag{1}$$

Unfortunately, this estimation is biased. The relation between the estimation and the true value can be found by

$$\begin{aligned}
U(\beta|v) &= -\mathbb{E}_{\pi(\theta|v,\beta)}[\log \pi(\beta|v)] \qquad (\text{since } \log \pi(\beta|v) \text{ is independent of } \theta) \\
&= -\mathbb{E}_{\pi(\theta|v,\beta)}[\log \pi(\beta, \theta|v) - \log \pi(\theta|v, \beta)] \\
&\approx -\frac{1}{N} \sum_{n=1}^N \left( \log \pi(\beta, \theta^{(n)}|v) - \log \pi(\theta^{(n)}|v, \beta) \right),
\end{aligned}$$

where an average of $-\log \pi(\theta^{(n)}|v, \beta)$ lies between them. Although $\log \pi(\theta|v, \beta)$ is known up to a shifting constant, it cannot meet the demand here since the $\theta$-free shifting constant varies with $\beta$, and we consider a function of $\beta$ here.

Nevertheless Eqn. (1) seems to be the only way to estimate the potential energy. It can be used by an approximation: when the proposal of $\beta$ is not distant from its beginning value, $\pi(\theta|v, \beta)$ does not change much. Adopting this, a sampling inference method for SAM by directly sample from $\pi(\beta|v)$ can be conceived, i.e. **GMC-apprMH**. The gradient is estimated in the same way as **SGGMC** except it uses the whole dataset.

Due to the inaccuracy of this, **GMC-bGibbs** is considered. But **GMC-apprMH** is in the same scheme of **SGGMC/gSGNHT** since they draw one sample of $\beta$ based on multiple samples of $\theta$, while **GMC-bGibbs** only use one sample of $\theta$ for each $\beta$.

## Appendix C: A Derivation of the Momentum and the Hamiltonian Originated from Physics

In classical mechanics (see e.g. [5]), there is a formal derivation for the Hamiltonian, beginning with the Lagrangian. In the coordinate space, the Lagrangian is defined by $L(q, \dot{q}) = \frac{1}{2}\dot{q}^\top G(q)\dot{q} - U_\mathcal{H}(\xi(q))$, where $q \in \Omega$ and $\dot{q} \in T_q\Omega = \mathbb{R}^m$. The Hamiltonian is defined as the Legendre transformation of the Lagrangian $L(q, \dot{q})$. To perform the transformation, define the (generalized conjugate) momentum as $p \triangleq \frac{\partial L}{\partial \dot{q}} = G(q)\dot{q}$, and accordingly express $\dot{q} = \dot{q}(q, p) = G(q)^{-1}p$, then the Hamiltonian is expressed as

$$H(z) \triangleq \left( p^\top \dot{q} - L(q, \dot{q}) \right)\Big|_{\dot{q}=\dot{q}(q,p)} = \frac{1}{2}p^\top G(q)^{-1}p + U_\mathcal{H}(\xi(q)),$$

where $z \triangleq (q, p)$ is called the canonical coordinates. This is the same as the one used by SGGMC.

In an isometrically embedded space, the same procedure can be applied. The Lagrangian now is $L(x, \dot{x}) = \frac{1}{2}\dot{x}^\top \dot{x} - U_{\mathcal{H}}(x)$, where $x \in \Xi(\mathcal{M})$ and $\dot{x} \in T_x \Xi(\mathcal{M})$ (the tangent space of $\Xi(\mathcal{M})$ at $x$). The momentum in the embedded space is thus $v \triangleq \frac{\partial L}{\partial \dot{x}} = \dot{x}$. The Hamiltonian is derived by

$$H(x, v) \triangleq \left( x^\top \dot{x} - L(x, \dot{x}) \right)\Big|_{\dot{x}=\dot{x}(x,v)} = \frac{1}{2}v^\top v + U_{\mathcal{H}}(x).$$

This is the common form of the Hamiltonian, where the "velocity" should be regarded as the momentum in the isometrically embedded space for a general case.

## Appendix D: A Rationale on the Shape of the Joint Posterior for the Synthetic Experiment

In this part we provide an interpretation for the symmetric bimodal shape of the posterior $\pi(v_1, v_2 | \mathcal{D})$ for the simple mixture of vMF model in Sec. 5.2 of the main paper. We start with the way we generate the synthetic data. They are samples drawn by GMC from the likelihood $\pi(x_i | v_1, v_2) \propto \text{vMF}(x_i | v_1, \kappa_x) + \text{vMF}(x_i | \mu, \kappa_x)$, with $\mu \triangleq (v_1 + v_2)/\|v_1 + v_2\|$, and $v_1 = v_1^{(g)}$ and $v_2 = v_2^{(g)}$ as shown in Fig. 1 (left). Due to the mono-modal shape of vMF, the synthetic data has two modes: $v_1^{(g)}$ and $\mu^{(g)}$. On the other hand, by referring to the generating process, the modes of $x$ are $v_1$ and $\mu$. Thus, we can approximately (under a weak prior) estimate the modes of the posterior by matching the theoretical data modes and the observed data modes: 1) let $v_1 = v_1^{(g)}, \mu = \mu^{(g)}$, we have $v_1^{(1)} = v_1^{(g)}, v_2^{(1)} = v_2^{(g)}$, i.e. the value used to generate

Figure 1: Left plot shows $v_1, v_2, \mu$ for generating data. Modes of the synthetic data are $v_1^{(g)}$ and $\mu^{(g)}$. Right plot shows the two approximate modes of the posterior under a weak prior: $(v_1^{(1)}, v_2^{(1)})$ in blue and $(v_1^{(2)}, v_2^{(2)})$ in green.

data; 2) let $v_1 = \mu^{(g)}, \mu = v_1^{(g)}$, we have $v_1^{(2)} = \mu^{(g)}$ and $v_2^{(2)}$ shown in Fig. 1 (right). Note that the two approximate modes of the posterior are symmetric with respect to $e_1$, so they share equal preference from the prior. In our settings $\phi = \frac{\pi}{6}$, so we expect the posterior to have two equal-weighted modes around $(-\frac{\pi}{24}, \frac{\pi}{8})$ and $(\frac{\pi}{24}, -\frac{\pi}{8})$. Our experiment results in the main paper meets this reasoning.

## Appendix E: Algorithms of SGGMC/gSGNHT and their application to SAM

Based on the statements in Sec. 3.3 of the main paper, we summarize the algorithms of our proposed SGGMC and gSGNHT in Alg. 1 and Alg. 2, respectively. Here we only present algorithms for *scalar* $C$.

For both SGGMC and gSGNHT, the step size schedule $\{\varepsilon_n\}$ is recommended to be a fixed number. Although a shrinking schedule, e.g. $\varepsilon_n \propto n^{-k}$ for $k \in (0, 1)$ mentioned in [3], enjoys more theoretical guarantees (sample average is asymptotically unbiased even if the stochastic gradient may not be unbiased [3]), but the advantage may not show up in practice and the performance may be affected by insufficient exploration.

The first two parameters of our SGGMC/gSGNHT, the fixed step size $\varepsilon$ and the scalar $C$, can be managed in a similar way of SGHMC [4], which utilizes its analogy to stochastic gradient descent (SGD) with momentum. By introducing two SGD terminologies, the per-batch learning rate $\gamma$ and the coefficient of momentum $\rho$ (both scalar), our parameters can be set as $\varepsilon = \sqrt{\gamma/|\mathcal{D}|}$ and $C = \rho/\varepsilon$. Typical values of $\gamma$ and $\rho$ are around 0.1 to 0.01. The last parameter is the number of steps $L$. For SG-MCMCs it is often set to 1, in the sense that $L$ does not affect the simulation trajectory, different from HMC. Other integers can also be used, with less correlated samples. Automatic selection of $L$ for SG-MCMCs remains to be studied, while the counterpart for HMC is provided as the No-U-Turn Sampler (NUTS) [6].

From the contents in Sec. 4 of the main paper, the algorithm to use SGGMC/gSGNHT for the inference problem of SAM is presented in Alg. 3.

---
**Algorithm 1** Sampling procedure of SGGMC
---
Randomly initialize $x^{(0)} \in \Xi(\mathcal{M})$.
Sample $v^* \sim \mathcal{N}(0, I)$ and project $v^{(0)} \leftarrow \Lambda(x^{(0)})v^*$.
**for** $n = 1, 2, \ldots,$ **do**
    Sample a subset $\mathcal{S}$ for computing $\nabla_x \tilde{U}_{\mathcal{H}}(x)$. $(x_0, v_0) \leftarrow (x^{(n-1)}, v^{(n-1)})$.
    **for** $l = 1, 2, \ldots, L$ **do**
        A: Update $(x^*, v^*) \leftarrow (x_{l-1}, v_{l-1})$ by the geodesic flow for time step $\frac{\varepsilon_n}{2}$.
        B: $v^* \leftarrow \exp\{-C\frac{\varepsilon_n}{2}\}v^*$.
        O: $v^* \leftarrow v^* + \Lambda(x^*) \cdot \left[ -\nabla_x \tilde{U}_{\mathcal{H}}(x^*)\varepsilon_n + \mathcal{N}\left(0, (2C - \varepsilon_n V(x^*))\varepsilon_n\right) \right]$.
        B: $v^* \leftarrow \exp\{-C\frac{\varepsilon_n}{2}\}v^*$.
        A: Update $(x_l, v_l) \leftarrow (x^*, v^*)$ by the geodesic flow for time step $\frac{\varepsilon_n}{2}$.
    **end for**
    $(x^{(n+1)}, v^{(n+1)}) \leftarrow (x_L, v_L)$. No M-H test.
**end for**
---

---
**Algorithm 2** Sampling procedure of gSGNHT
---
Randomly initialize $x^{(0)} \in \Xi(\mathcal{M})$.
Sample $v^* \sim \mathcal{N}(0, I)$ and project $v^{(0)} \leftarrow \Lambda(x^{(0)})v^*$. $\xi^{(0)} \leftarrow C$.
**for** $n = 1, 2, \ldots,$ **do**
    Sample a subset $\mathcal{S}$ for computing $\nabla_x \tilde{U}_{\mathcal{H}}(x)$. $(x_0, v_0, \xi_0) \leftarrow (x^{(n-1)}, v^{(n-1)}, \xi^{(n-1)})$.
    **for** $l = 1, 2, \ldots, L$ **do**
        A: Update $(x^*, v^*) \leftarrow (x_{l-1}, v_{l-1})$ by the geodesic flow for time step $\frac{\varepsilon_n}{2}$,
           $\xi^* \leftarrow \xi_{l-1} + (\frac{1}{m} v_{l-1}^\top v_{l-1} - 1)\frac{\varepsilon_n}{2}$.
        B: $v^* \leftarrow \exp\{-\xi^* \frac{\varepsilon_n}{2}\}v^*$.
        O: $v^* \leftarrow v^* + \Lambda(x^*) \cdot \left[ -\nabla_x \tilde{U}_{\mathcal{H}}(x^*)\varepsilon_n + \mathcal{N}\left(0, (2C - \varepsilon_n V(x^*))\varepsilon_n\right) \right]$.
        B: $v^* \leftarrow \exp\{-\xi^* \frac{\varepsilon_n}{2}\}v^*$.
        A: Update $(x_l, v_l) \leftarrow (x^*, v^*)$ by the geodesic flow for time step $\frac{\varepsilon_n}{2}$,
           $\xi_l \leftarrow \xi^* + (\frac{1}{m} v^{*\top} v^* - 1)\frac{\varepsilon_n}{2}$.
    **end for**
    $(x^{(n)}, v^{(n)}, \xi^{(n)}) \leftarrow (x_L, v_L, \xi_L)$. No M-H test.
**end for**
---

## Appendix F: Implementation Details for Experiments

### F.1: Toy experiment in Sec. 5.1.

To draw 10,000 samples by each method, we set $L = 30$, $\varepsilon = 1 \times 10^{-2}$ for both GMC and SGGMC, and $\rho = 0.1$ for SGGMC. For the empirical distribution, the bin size is set to 0.1.

### F.2: Synthetic experiment in Sec. 5.2.

The 100-sized synthetic data is generated by GMC without burn-in. To draw from the posterior, we set $L = 20$ and $\varepsilon = 1 \times 10^{-3}$ for GMC, and $L = 10$, $\rho = 0.1$ and $\gamma = 5 \times 10^{-4}$ for SGGMC. For each method, 25,000 posterior samples are taken after 15,000 burned in.

### F.3: Spherical admixture model experiment in Sec. 5.3.

The two datasets and codes for all inference methods are available at `http://ml.cs.tsinghua.edu.cn/~changliu/sggmcmc-sam/`.

**Datasets** Both datasets present the *term frequency-inverse document frequency*, or *tf-idf* feature of documents. They are converted from the *bag-of-words* feature of their corresponding original datasets, which provide *tf*$(d, v)$ — term frequency (number of occurrence) of term $v$ in document $d$. The conversion is done by first computing *tf-idf*$(d, v) = $ *tf*$(d, v) \cdot \log\left(D/(1 + df(v))\right)$, where $D$ is the number of documents in the dataset, and *df*$(v)$ is the document frequency of term $v$ (number of

---
**Algorithm 3** sampling inference for SAM using SGGMC/gSGNHT
---
    **for** $m = 1, 2, \ldots$ **do**
        Randomly sample a subset $\{d(s)\}_{s=1}^{S}$ from whole training data.
        **for** $s = 1, 2, \ldots, S$ **do**
            Sample $N$ times from $\pi(\theta_{d(s)}|\beta^{(m-1)}, v_{d(s)})$ using GMC to get $\{\theta_{d(s)}^{(n)}\}_{n=1}^{N}$.
        **end for**
        Sample once from $\pi(\beta|v)$ using SGGMC/gSGNHT to get $\beta^{(m)}$, with stochastic gradient computed by Eqn. (10) in the main paper.
    **end for**
---

documents containing term $v$), then $\ell_2-$normalizing the vector with component $v$ equal to *tf-idf*$(d, v)$ for a fixed $d$ and getting the unit vector *tf-idf*$(d)$ for document $d$.

The small dataset 20News-different is a subset of the 20Newsgroups dataset (`http://www.qwone.com/~jason/20Newsgroups/`, we use the Matlab/Octave version). It contains 3 categories out of the total 20: *rec.sport.baseball*, *sci.space* and *alt.atheism*. It is used by [8] to illustrate the benefits of SAM over LDA [1] in feature reduction. The original vocabulary size is 61,188, and we shrink it to 5,000 by selecting words with moderate document frequency (between 0.36% and 11.77%). While training by any method on the dataset, hyper-parameters of SAM are fixed as $\sigma = 1 \times 10^4$, $\kappa_0 = 1 \times 10^4$, $\kappa_1 = 3 \times 10^4$, $\alpha = 10$, and $m$ set to the normalized mean of training documents.

The large dataset is based on the 6.6M Wikipedia dataset used by [9] (`http://ml.cs.tsinghua.edu.cn/~aonan/datasets/wikipedia/`). The original vocabulary size is 7,702, and we shrink it to 3,000 by selecting words with moderate document frequency (between 0.44% and 5.99%). Our dataset is then generated by randomly (excluding documents with $\leq 20$ words) choosing 150K training and 1K test documents from the 6.6M whole dataset. The training size is the same as used by [7] for presenting scalability. While training by any method on the dataset, hyper-parameters of SAM are fixed as $\sigma = 6 \times 10^3$, $\kappa_0 = 6 \times 10^3$, $\kappa_1 = 2 \times 10^4$, $\alpha = 10$, and $m$ set to the normalized mean of training documents.

**Issues on the sampling methods** For all sampling methods, samples of topic proportion $\theta_d$ of document $d$ drawn from $\pi(\theta_d|v_d, \beta)$ are required. This can be well done by GMC. We use the initialization $\theta_d = (\beta^\top \beta)^{-1}\beta^\top v_d$ for sampling $\theta_d$, which is the mode of $\pi(\theta_d|v_d, \beta)$ under an uninformative prior $\alpha = 1$. For **GMC-apprMH/GMC-bGibbs**, to draw one sample of $\beta$, sampling $\theta_d$ are carried out for all the documents in the training data, while **SGGMC/gSGNHT** only need to traverse the chosen mini-batch. Noting that drawing $\theta_d$ for different $d$ is independent of each other, we parallelize the sampling procedure for different $d$, by OpenMP (`http://openmp.org/`). Since all sampling methods involve this and once for one $\beta$ sample, it is still fair to compare the methods by the evolution along wall time, as long as the number of threads is the same.

**Techniques to avoid overflow** We need further special techniques to avoid the numerical overflow problem, which is caused by the modified Bessel function of the first kind $I_r(\cdot)$ in order $r$ that is used in the normalization constant of vMF distribution (See Sec. 4 of the main paper). $I_r(\cdot)$ tends to be either zero or infinity on sides of some argument threshold when the order $r$ is large. In our experiments $r = V/2 - 1$ is of order of thousands, and the behavior is obvious. To avoid trivial models, we have to choose hyper-parameters of SAM $\sigma$, $\kappa_0$ and $\kappa_1$ (acting as arguments of $I_r(\cdot)$) relatively large so that $I_r$ is non-zero. But it then almost always overflows. Thanks to the fact that only its logarithm is required, we can try to directly calculate the logarithm. By noting that

$$\log I_r(x) = \log \left( \sum_{n=0}^{\infty} \frac{1}{n!\Gamma(n+r+1)} \left(\frac{x}{2}\right)^{2n+r} \right)$$

is the logarithm of a summation, we can use the log-sum trick: to calculate $\log(A + B)$ with only $a = \log A$ and $b = \log B$ (assume $a \geq b$) available, we can reformulate the target as

$$\log(A + B) = \log(\exp(a) + \exp(b)) = \log\big(\exp(a)\big(1 + \exp(b - a)\big)\big) = a + \log\big(1 + \exp(b - a)\big),$$

where $b - a \leq 0$ thus $1 + \exp(b - a) \leq 2$, so no numerical instability is met.

For specific implementation and parameters of our methods for the presented results, please refer to our codes.