[Reviews · NeurIPS 2016]

Reviewer 1

Summary

This paper extends the geodesic MCMC algorithm to a stochastic gradient geodesic MCMC (sggmcmc) algorithm. The proposed method is a straight application of the method proposed in Ma et. al. 2015, which is rather incremental.

Qualitative Assessment

This paper is hard to read. This paper extends the geodesic MCMC algorithm to a stochastic gradient geodesic MCMC (sggmcmc) algorithm. The proposed method is a straight application of the method proposed in Ma et. al. 2015, which is rather incremental. Similar to the stochastic gradient MCMC methods, the sggmcmc also lacks a Metropolis-Hastings correction step.

Confidence in this Review

2-Confident (read it all; understood it all reasonably well)


Reviewer 2

Summary

This paper exploits the geodesic flow to perform stochastic MCMC. The authors leverage the symmetric splitting integrator for the numerical simulation. The two proposed methods are evaluated by synthetic and real data.

Qualitative Assessment

1. The extension of SGGMC from previous work (SGRHMC)[1] are in two folds. First, the proposed method use Geodesic flow rather than Riemmannian manifold. Second, the proposed method leverage a symmetric splitting integrator (ABOBA) scheme. However, unfortunately none of extensions have a clear and convincing novelty as far as I can see. The drop-in replacement of D and Q are not surprising. The main difference comparing to SGRHMC is to replace the fisher information $G(q)$ with geodesic metric $M(q)^T C M(q)$, thus eliminating the need for global coordinates, which may induce artificial boundaries in some manifolds [4]. This is well-studied in [4], thus the contribution seems marginal. The SSI scheme is originally proposed in [2], and what they did seems to be a direct application of the standard SSI. In fact as pointed out by [3], it does not lead to correct sampler if SSI is not applied, as the motion is not reversible and symplectic. The gSGNHT is not very appealing either because its an immediate extension from [1]. I am expecting to see some cases where the SGRHMC fails (e.g. due to a lack of global coordinates), and their methods survives. However, in their experiments, they didn't perform any comparison between their proposed method with [1]. Thereby I am not convinced that they did provide something more useful than [1]. 2. In GMC[4] the splitting integrator is guaranteed to be reversible and symplectic. However this may not directly applied to cases where stochastic gradient is used. I haven't found any clear theoretical evidences in the paper supporting that their splitting integrator is still reversible and symplectic. 3. The toy and synthetic experiments does not provide insight on why SGGMC is more desirable than GMC. The results of SGGMC seems suboptimal, particularly in Fig. 3(b) and Fig. 4(a). A comparison on running time would help. There is no comparison with SGRHMC. 4. In real data experiment, the compared sampling methods does not seem to reach a convergence. I would provide a better intuition on why the stochastic version gives a very large performance gain on log-perplexity (from 3000 to 1500 in Fig. 5(a)). I would also provide more information about the parameters (e.g. stepsize) and initialization setting, otherwise it is hard to conclude where the benefit comes from. The paper is well written and pedagogical, though more introductions about Geodesic flow would be helpful. [1] Yi-An Ma, Tianqi Chen, and Emily Fox. A complete recipe for stochastic gradient mcmc. [2] Changyou Chen, Nan Ding, and Lawrence Carin. On the convergence of stochastic gradient mcmc algorithms with high-order integrators. [3] Mark Girolami and Ben Calderhead. Riemann manifold langevin and hamiltonian monte carlo methods. [4] Simon Byrne and Mark Girolami. Geodesic monte carlo on embedded manifolds.

Confidence in this Review

2-Confident (read it all; understood it all reasonably well)


Reviewer 3

Summary

This paper proposes two novel SG-MCMC methods for generating samples from densities defined on Riemannian manifolds. One of the methods can be seen as an extension of GMC to the stochastic gradient setting, whereas the second method is based on a more sophisticated dynamics. The methods are illustrated on a challenging problem.

Qualitative Assessment

The topic of the paper is very interesting. The paper is well written, I enjoyed reading it. Apart from some minor comments detailed below, I only have the following concerns: * Even though the proposed methods can handle complicated densities with known geodesic flows, I do not think the methods have a wide application area. The authors illustrate their methods on a SAM model but I am not sure if these methods would be more advantageous than the existing methods on other problems. * The motivation parts of the paper can be improved. The authors use differential geometric terms starting from the abstract. For a general reader from MCMC domain might not be aware of these constructs and the motivations of the paper might be unclear. On the other hand, some other parts of the paper also needs more clear introduction. For instance, the reason why the authors directly develop a second order integrator is not well-motivated. They mention that they aim to circumvent the usage of inner iterations by using the second order integrator but all these things are explained in a rather implicit way. * This paper (see below) is very relevant. The authors should discuss the differences between their methods and this paper. Roychowdhury et al., Robust Monte Carlo Sampling using Riemannian Nosé-Poincaré Hamiltonian Dynamics, ICML 2016 == Minor comments == * There are also other relevant studies that can be cited, such as Zhang and Sutton, Quasi-Newton Methods for Markov Chain Monte Carlo, NIPS 2011 Li et al., Preconditioned Stochastic Gradient Langevin Dynamics for Deep Neural Networks, AAAI 2016 Simsekli et al., Stochastic Quasi-Newton Langevin Monte Carlo, ICML 2016 Fu et al., Quasi-Newton Hamiltonian Monte Carlo, UAI 2016 * Line 58: brief -> briefly * Line 62: gradient -> the gradient * Line 63: The definition of U(q) is not correct. U(q) = - log pi(q) - \sum_d log pi(x_d|q). But -log pi(q|D) is not equal to (but proportional to) - log pi(q) - \sum_d log pi(x_d|q). * Line 106 (and others): pi(q) is first defined as the prior distribution, but then it is used as the target distribution. * Line 107: H^m should be H^n * Line 133: More explanation is needed on why the integrator in GCM "does not suit". * Figure 5: wall time -> wall-clock time

Confidence in this Review

3-Expert (read the paper in detail, know the area, quite certain of my opinion)


Reviewer 4

Summary

This paper proposes two stochastic gradient MCMC methods for Riemannian manifolds with tractable geodesic flows, and demonstrates their performance on several toy experiments, and a large-scale real data experiment.

Qualitative Assessment

The overall structure of the paper is reasonably clear, although I think the clarity suffers from not having explicit statements of the proposed algorithms in the main text. With regards to technical quality, the introductory exposition is a little loose in some places e.g. Eqn (1) states that the stochastic gradient noise is exactly normal, and line 81 claims that the dynamics can be exactly achieved with stochastic gradients - both of these claims rely on some kind of asymptotic approximation. Equation (4) is unclear to me - it is unclear what dt^2 is intended to mean, and where it has come from. The motivation for the exact forms of splittings used in constructing the numerical integrators (e.g. line 175-177) for the two algorithms is not clear to me - is there a reason for selecting an ABOBA splitting in particular, either from a theoretical perspective, or empirically? There is also no reference provided for the mentions of “SSI” in the paper, and it is not clear whether the authors are referring generally to integration schemes based on symmetric splittings, or to some more specific integrator. With regards to the toy/synthetic experiments, these are both carried out on manifolds with a global isometric parametrization (the torus T^2, specifically), which presumably is amenable to stochastic gradient methods already proposed in the literature such as SGRLD etc. - it would be interesting to see these existing methods applied to the toy problems for comparison. On a similar note, it would be useful to see a toy example on a manifold that doesn’t have a global isometric parametrization (e.g. S^2). The conclusion “we conclude that SGGMC is valid” is perhaps a bit strong to draw from the toy experiment in Section 5, and it is not clear what precisely is meant. The performance on the spherical admixture model is impressive, and is a strong use case for the algorithms. Generally, although the toy experiments show sensible results and the real-data experiment shows good performance, there is little theoretical support for the algorithms presented in the paper, and I believe there are several important questions that could be addressed more directly in the paper, such as i) how does ignoring the MH step affect the algorithms? ii) what numerical errors can be expected to be incurred by the discretisation? iii) what step-size regimes were used in the experiments/are recommended for the algorithms? Minor points: Line 15: manifold -> manifolds Line 65: the gradient of the log prior is missing a minus sign. Line 95: The map \Phi should be a diffeomorphism, not just a homeomorphism Line 112: an Euclidean -> a Euclidean Throughout: The acronym SSI is used several times before it is defined

Confidence in this Review

2-Confident (read it all; understood it all reasonably well)


Reviewer 5

Summary

The authors develop two flavors of stochastic gradient samplers suitable for Riemannian manifolds. The numerical evidence shows that the methodology works well for the spherical admixture models.

Qualitative Assessment

The main problem of the paper is that it too dense and that it becomes very hard to read at certain points. Also, there are several typos and grammatical mistakes spread all over the text. I am not going to enumerate those. I'm sure that the authors will take care of them. Please consider responding to following comments/suggestions: 1. The paper is very difficult to read. I believe that you would have a better chance of highlighting your contributions if you just presented one of your samplers. 2. Eq (1). First, $\mathcal{N}(0, V(q))$ is not defined, even though we can all guess what it is. Second, as a mathematician, I do not like the fact that you represent a random variable with its law. The notation seems very strange to me. I would rather see this: $$ \nabla_q\tilde{U}(q) = \nabla_q U(q) + \epsilon, $$ where $\epsilon \sim \mathcal{N}(0, V(q))$, or something similar. Of course, this applies to many other parts of the text. 3. Page 2. About the work of Ma et al. You state that "It claims that for ...". Is it really a "claim" or is it a "proof"? 4. Page 5, line 182. Doesn't the "tf-idf representation of text data" require a reference? 5. All figures. The fonts of legends and labels are just too small.

Confidence in this Review

2-Confident (read it all; understood it all reasonably well)


Reviewer 6

Summary

This paper generalises SGHMC and SGNHT from the Euclidian Space R^n to manifolds with a known geodesic flow. The novel algorithms, which are the first ones to enable scalable inference on these manifolds, do not require the computationally expensive inner iteration nor a global coordinate system. In addition, the authors develop second-order integrators to further improve on performance. An application to the problem of spherical admixture models is presented. Both synthetic and real-world data experiments confirm the validity of the new approach, which exhibits better accuracy and scales better than competing methods.

Qualitative Assessment

The paper writing is clear and accurate (despite some minor grammar flows, such as articles missing). The authors give a good background prior to explaining the main ideas, which makes the paper relatively easy to follow. Even though the Ma/Chen/Fox framework makes the derivation of the new algorithms pretty straightforward, it was interesting to see that the authors developed a second-rate geodesic integrator for the new methods and detailed its application to the SAM. Overall a very interesting contribution and well-presented paper.

Confidence in this Review

2-Confident (read it all; understood it all reasonably well)